# APPROXIMATE MULTI-MATRIX MULTIPLICATION FOR STREAMING POWER ITERATION CLUSTERING

## ABSTRACT

Given a graph, accurately and efficiently detecting the communities present is one of the main challenges in network analysis. In this era, where datasets routinely exceed terabytes in size, many classical algorithms for solving this problem become computationally prohibitive. We address this challenge in the context of the Stochastic Block Model (SBM), which allows for a rigorous analysis. Our approach is a sublinear, updateable, and single-pass approximation to a classic power iteration algorithm (Mukherjee & Zhang, 2024). We introduce two sketching-based variants: (1) a *streaming algorithm* for single-pass processing of edge streams, and (2) an *r-pass algorithm* that achieves a smaller space embedding at the cost of additional passes equal to the power $r$ of the matrix to be approximated. We show that both methods produce vertex embeddings that guarantee the recovery of the largest cluster when performing single-linkage clustering with an appropriate *separation scale* cut threshold.

Our key contribution is a new theoretical analysis of Approximate Multi-Matrix Multiplication (AMMM), which guarantees that the error from repeated compression remains manageable. This framework extends the stable-rank-based approximate matrix multiplication (AMM) guarantees of (Cohen et al., 2016) to arbitrarily many conforming matrices. We prove that both algorithms preserve the geometric structure needed to identify the largest community using sublinear space in practice. The streaming algorithm (1) scales with the stable rank of the graph matrix for the streaming algorithm, which we show is sublinear in practice. The $r$-pass algorithm achieves the optimal $O(\varepsilon^{-2} \log n)$. Experiments on synthetic graphs confirm that our methods can recover the largest community as effectively as the exact, expensive algorithm, across both balanced and unbalanced communities, but with dramatically lower memory and runtime.

## 1 INTRODUCTION

Community detection in graphs is a fundamental problem with applications across network analysis, machine learning, and data mining. In this work we focus on spectral methods that learn low-dimensional vertex representations via algebraic measurements of graph matrices. Although these methods remain popular in practice, they face significant scalability challenges in modern datasets, which can be large, evolving, or both. Conventional spectral embedding methods utilize iterative graph decompositions or power iteration stages, requiring access to the whole graph matrix. Furthermore, these embeddings are often difficult to update if the graph evolves, even slightly.

In this work we seek to overcome these challenges by designing a turnstile streaming embedding. A turnstile (sometimes called dynamic) streaming algorithm is robust to the addition or deletion of edges, which is essential for evolving or temporal data. Moreover, we seek an embedding that is strongly sublinear in the size of the graph matrix, which is critical when the graph is very large. Our investigation yields a trade-off between pass complexity and embedding size, so in addition to a single-pass streaming algorithm we also analyze an $r$-pass variant with smaller space.

**Our Contribution.** We introduce and analyze two efficient variants of power iteration clustering for the stochastic block model (SBM), both building on the foundation laid by Mukherjee & Zhang (2024). Their work established that power iteration of a centered adjacency matrix $B = A -$

$q\mathbf{1}\mathbf{1}^\top$ produces row embeddings where the largest community becomes separable by a *separation scale* threshold. Rather than computing $B^r$, our algorithms approximate this power iteration using Johnson-Lindenstrauss transform sketches. Our contributions are:

- **Streaming Algorithm:** We design a single-pass algorithm for dynamic edge streams that interleaves randomized sketches between matrix multiplications, computing $\widetilde{B}^{(r)} = BS_1^\top S_1 BS_2^\top \ldots S_{r-2} BS_{r-1}^\top$ using appropriately sampled Johnson-Lindenstrauss transform matrices $S_1, S_2, \ldots, S_{r-1}$. This enables processing graphs too large for memory while using sublinear embedding dimensions. Moreover, the sketch factors $BS_1^\top$ and $S_i BS_{i+1}^\top$ are updatable with both edge insertions and deletions.

- $r$**-pass Algorithm:** We also analyze an $r$-pass variant $B^r S$ that achieves smaller space at the cost of additional passes a loss of robustness to changes in $B$.

- **Theoretical Framework:** We develop *Approximate Multi-Matrix Multiplication (AMMM)*, extending the stable-rank optimal sketching of Cohen et al. (2016) to matrix chains. For any accuracy parameter $\varepsilon \in (0, 1)$ controlling the relative error in matrix product approximations, AMMM provides non-trivial guarantees that error accumulates controllably across additional matrices.

- **Recovery Guarantees:** We prove both algorithms preserve the separation scale needed for largest-community recovery, with explicit embedding dimensions: $\tilde{O}(\text{sr}(B)\varepsilon^{-2})$ for streaming and $O(\varepsilon^{-2} \log n)$ for the stored-matrix setting. It is useful to note that $\text{sr}(B)$ denote the stable rank of $B$ while $\epsilon \in (0, 1)$ is the distortion level introduced by the sketch matrices used.

Our AMMM analysis reveals a crucial *multi-step stability property*: the perturbation from sketching grows linearly rather than exponentially with iterations. This enables our single-pass streaming approach, where each sketch can be accumulated independently and combined with a single reduction thereby making the algorithm practical for distributed and streaming environments.

Empirically, we validate that both variants recover the largest community in SBMs while preserving the theoretical separation scale. We show that this preservation occurs both in the balanced regime where community sizes are roughly uniform and the unbalanced regimes, where community sizes are skewed and there are many small communities. The streaming method achieves this with slightly larger embedding dimensions but requires only one pass, representing a fundamental trade-off between memory and communication efficiency.

## 2 NOTATIONS, MODEL, AND GOAL

In this section, we formalize our compute model and objectives. A graph $G = (V, E)$ is drawn from an SBM if it comes from the distribution

$$G \sim \text{SBM}\big(n, K, \{s_\ell\}_{\ell=1}^K, p, q\big), \qquad 0 < q < p < 1,$$

where $n$ is the number of vertices, $K$ is the number of communities (clusters), and $s_\ell$ is the size of community $\ell$ with $\sum_{\ell=1}^K s_\ell = n$ (we allow unbalanced communities), with $p$ and $q$ defined below.

For a positive integer $x$, define $[x] = \{1, 2, \ldots, x\}$. We refer to the vertices by integer indices, so $V = [n]$. Let $C_\ell \subseteq [n]$ and $|C_\ell| = s_\ell$ denoted the index set and cardinality of community $\ell$ respectively. The communities are disjoint, so $V = \bigcup_{\ell=1}^K C_\ell$ and $C_\ell \cap C_{\ell'} = \emptyset$ for all $\ell \neq \ell'$. We write $s_* = \max_{\ell \in [K]} s_\ell$ and $s_{\min} = \min_{\ell \in [K]} s_\ell$. We correspondingly denote the largest and smallest communities with $C_*$ and $C_{\min}$, respectively. For vertices $i$ and $j$, we use the notation $i \sim j$ when $i, j \in C_\ell$ for some $\ell$ and $i \nsim j$ if $i \in C_\ell$ and $j \in C_{\ell'}$, where $\ell \neq \ell'$. So, $p$ and $q$ represent the probability of an edge connecting $i$ and $j$ when $i \sim j$ and $i \nsim j$, respectively. The adjacency matrix $A \in \{0, 1\}^{n \times n}$ is symmetric and, for $i, j \in [n]$,

$$A_{ij} \sim \begin{cases} \text{Bernoulli}(p), & \text{if } i \sim j \\ \text{Bernoulli}(q), & \text{if } i \nsim j. \end{cases}$$

Throughout, we write $\| \cdot \|_{\text{op}}$ for the operator norm, which coincides with the spectral norm in our setting, and $\| \cdot \|_{\text{F}}$ for the Frobenius norm of matrices; the (matrix) stable rank is $\text{sr}(X) := \|X\|_{\text{F}}^2 / \|X\|_{op}^2$. We use $\| \cdot \|$ to refer generically to any submultiplicative matrix norm. We write $\| \cdot \|_2$ for the usual $\ell_2$-norm of vectors.

**Centered model.** Following the power–iteration analysis in Mukherjee & Zhang (2024), we work with the $q$–centered matrix as defined in their paper:

$$B := A - q\mathbf{1}\mathbf{1}^\top = \boldsymbol{L} + R, \qquad L := \mathbb{E}[B], \quad R := B - L. \tag{1}$$

Informally, $L$ is the *signal* part: a deterministic rank–$K$ block structure that encodes the community means relative to the baseline level $q$. Mukherjee & Zhang (2024) show that $q$ can be estimated with $\frac{d_{\min}}{n}$, where $d_{\min}$ is the smallest degree in $G$ (alternately, the smallest support of a row in $A$). The term $R$ is the *noise* (mean–zero fluctuation) around that signal: it is symmetric and has independent upper–triangular entries. The centered matrix $B$ is created specifically so the elements of $R$ have zero mean and variances $\leq p(1-q)$. Lemma 2.1 of Mukherjee & Zhang (2024) shows there exists an absolute constant $C_0 > 0$ such that $\|R\|_{\mathrm{op}} \leq C_0 \sqrt{p(1-q)} \sqrt{n}$ with probability at least $1 - n^{-3}$.

**Goal: largest–cluster recovery.** Our target is to recover the index set $C_*$ of the largest community. In the exact algorithm considered in Mukherjee & Zhang (2024), one forms *power features* by taking rows of $B^r$ (for a modest power $r$, e.g. $r = O(\log n)$ or a small fixed $r$) and clusters these row vectors via single linkage clustering with a separation scale threshold $\Delta_r$. The correctness analysis in Mukherjee & Zhang (2024) is phrased in terms of a $\Delta_r$: there exist thresholds $\tau_{\mathrm{in}}, \tau_{\mathrm{out}}$ with margin $\gamma := \tau_{\mathrm{out}} - \tau_{\mathrm{in}} > 0$, such that the within–$C_*$ pairwise row distances of $B^r$ are at most $\tau_{\mathrm{in}}$ and the across–$C_*$ vs. $[n]\backslash C_*$ distances are at least $\tau_{\mathrm{out}}$.

We focus on the setting where $A$ is too large to store and may arrive as a dynamic (turnstile) stream of edge insertions and deletions. We therefore interleave the powers of $B$ with oblivious sketches $S_1, \ldots, S_r \in \mathbb{R}^{m \times n}$ in the form $\widetilde{B}^{(r)} = B S_1^\top S_1 B S_2^\top S_2 \cdots B S_{r-1}^\top S_{r-1} B S_r^\top$. Note that the linearity of the random projections implies that the individual factors $BS_1^\top$ and $S_i B S_{i+1}^\top$ can be individually updated with edge insertions and deletions and only multiplied together at clustering time. Linearity also allows us to apply the $q\mathbf{1}\mathbf{1}^\top$ component of $B$ to the sketches at multiplication time. Appendix B provides more detail.

Our AMMM analysis in §4 and §5 shows that, whenever the power features $B^r$ satisfy the separation scale $\Delta_r$ a *similar* threshold works for $\tilde{B}^r$, provided the embedding dimension $m = \tilde{O}(\mathrm{sr}(B)\,\varepsilon^{-2})$ where $\epsilon \in (0,1)$ is the distortion introduced by the sketches. We also show a simple analysis in §5.3 of a simpler $r$-pass algorithm. We compute $B^r S^\top = B(B(\ldots(BS^\top)\ldots))$ using a single sketch $S \in \mathbb{R}^{m \times n}$ in $r$ passes by multiplying the intermediate product by $B$ on the left in each pass. This is similar to the algorithms considered by Zhang et al. (2018) and Macgregor (2023). Note that we can apply the rank-one term, $q\mathbf{1}\mathbf{1}^\top$, during each of these multiplications, avoiding the need to store or multiply using the dense $B$ matrix. A standard Johnson-Lindenstrauss argument shows multiplicative preservation of all row distances for $m = O(\varepsilon^{-2} \log n)$, so the separation scale scheme succeeds.

## 3 EXACT POWER ITERATION, ROW DISTANCES, AND THE SEPARATION SCALE

We begin by recalling the structure of $L$ and the separation scale, borrowing from the analysis in Mukherjee & Zhang (2024). The following is a self-contained derivation.

**Lemma 3.1** (Norms of $L$ and row separation). *We have* $\|L\|_{\mathrm{op}} = (p-q)s_*$ *and* $\|L\|_{\mathrm{F}}^2 = (p-q)^2 \sum_{\ell=1}^K s_\ell^2$. *Moreover, rows of $L^r$ are constant within each cluster and, if $i \in C_\ell$ and $j \in C_{\ell'}$ with $\ell \neq \ell'$,*

$$\|L_{i,\cdot}^r - L_{j,\cdot}^r\|_2 = (p-q)^r \sqrt{s_\ell^{2r-1} + s_{\ell'}^{2r-1}}. \tag{2}$$

*Proof.* Deferred to Appendix C. $\square$

We define the separation scale as

$$\Delta_r := (p-q)^r s_*^{r-\frac{1}{2}}. \tag{3}$$

Theorem 1.1 of Mukherjee & Zhang (2024) asserts that, if the largest community size $s_*$ and the separation scale $\Delta_r$ are large enough relative to $p$ and $q$, then the gap between the rows of $B^r$ are large enough for single linkage clustering to separate the largest cluster.

**Theorem 3.1** (Recovering largest cluster (Theorem 1.1 of Mukherjee & Zhang (2024))). *There are constants $C, C_0 > 0$ such that the following holds. Let $p, q \leq 0.75$ be parameters such that $\max\{p(1-p), q(1-q)\} \geq C_0(\log n)/n$. Let $G$ be a random graph sampled from $\mathrm{SBM}(n, K, \{s_\ell\}_{\ell=1}^K, p, q)$, and let $s_*$ be the size of the largest cluster.[1]If $s_* \geq C \cdot (\log n)^7 \cdot \sqrt{p(1-q)} \cdot \sqrt{n}/(p-q)$, then with probability $1 - O(1/n)$ one of the clusters produced by Algorithm 1 is the largest cluster of $G$ for $\Delta_r = \frac{1}{2}\sqrt{s^*}(p-q)^r(s^*)^{r-1}$ and $r = \lceil \log n \rceil$.*

Our proofs use the following assumption that naturally follows given the conditions of Theorem 3.1.

**Assumption 3.2** (Unsketched separation scale). *There exist $0 < a < b$, $\eta > 0$ very small and power $r$ such that with probability at least $1 - \eta$ over the graph,*

$$i \sim j \Rightarrow \|B_{i,\cdot}^r - B_{j,\cdot}^r\|_2 \leq a\,\Delta_r, \qquad i \not\sim j \Rightarrow \|B_{i,\cdot}^r - B_{j,\cdot}^r\|_2 \geq b\,\Delta_r.$$

We are now able present our algorithmic framework. Algorithm 1 represents our chassis for all algorithms considered, as they vary only in the implementation of the embedding subroutine `PI_SUBROUTINE`. Algorithm 1.1 in Mukherjee & Zhang (2024) corresponds to the clustering Algorithm 1 using Subroutine A as `PI_SUBROUTINE`. We will abuse notation and refer to this composition as Algorithm A for brevity, and similarly to the streaming and $r$-pass variants as Algorithm B and Algorithm C, respectively. Appendix B explains the streaming update behavior of SubroutineB in more explicit terms. The correctness of Algorithm A is guaranteed by Theorem 3.1 (Theorem 1.1 of Mukherjee & Zhang (2024)), while we will show the correctness of Algorithms B and C.

---

**Algorithm 1** DETECTING COMMUNITIES BY POWER ITERATION FRAMEWORK

---

1: **Input:** A graph $G$ and parameters $p, q > 0$.
2: Let $A$ be the adjacent matrix of $G$
3: $B \leftarrow A - q\mathbf{1}\mathbf{1}^\top$
          # (Here $\mathbf{1}^{n \times n}$ is the all 1 matrix.)
4: Let $\Delta_r > 0$ and $r > 1$ be parameters    # (We will explain how to choose them later)
5: $R^{(r)} \leftarrow \mathrm{PI\_SUBROUTINE}(B, r)$
6: **for** $v_i, v_j \in G$ **do**
7:   **if** $\|R_i^{(r)} - R_j^{(r)}\|_2 \leq \Delta_r$ **then**
8:     Put $v_i$ and $v_j$ in a same cluster    # ($R_i^{(r)}$ represents the $i$-th row of $R^{(r)}$)
9:   **end if**
10: **end for**
11: Output the sets thus formed.

---

| **Subroutine A** Exact PI | **Subroutine B** Streaming PI | **Subroutine C** $r$-pass PI |
|---|---|---|
| 1: **Input:** $B, r > 1$ 
 2: Output $B^r$ | 1: **Input:** $B, r > 1$ 
 2: Sample $S_1, \ldots, S_{r-1} \in \mathbb{R}^{m \times n}$ 
 3: Output $BS_1^\top \prod_{i=1}^{r-1} S_i B S_{i+1}^\top$ | 1: **Input:** $B, r > 1$ 
 2: Sample $S \in \mathbb{R}^{m \times n}$ 
 3: Output $B^r S^\top$ |

Armed thus with the prior work of Mukherjee & Zhang (2024), we are ready to proceed to prove largest cluster guarantees for Algorithm B.

## 4 APPROXIMATE MULTI-MATRIX MULTIPLICATION (AMMM)

This section extends *approximate matrix multiplication* (AMM) from a pair of products to *compositions of linear operators* with sketches inserted between consecutive factors. This framework arises naturally in iterative methods like power iteration, where we aim to replace exact matrix powers with sketched approximations while preserving downstream geometric guarantees. Our results hold for any *submultiplicative* matrix norm $\|\cdot\|$ (e.g., spectral or Frobenius) with AMM guarantees. Proof details appear in the appendix; here we provide a self-contained summary sufficient for later applications.

---

[1]Our parameterization of SBM differs from Mukherjee & Zhang (2024), who ignore the cluster sizes.

## 4.1 SETUP AND ONE-STEP AMM

Let $A_1, \ldots, A_k \in \mathbb{R}^{n \times n}$ be matrices and $S_1, \ldots, S_{k-1} \in \mathbb{R}^{m \times n}$ be independent sketching matrices. We compare the exact product $M_k := A_1 A_2 \cdots A_k$ with its sketched counterpart:

$$\widetilde{M}_k := A_1 S_1^\top S_1 A_2 S_2^\top S_2 \cdots A_{k-1} S_{k-1}^\top S_{k-1} A_k. \tag{4}$$

**Definition 4.1** (One-Step $(\varepsilon, \delta)$ AMM). *A matrix $S \in \mathbb{R}^{m \times n}$ satisfies one-step AMM at accuracy $(\varepsilon, \delta)$ for a norm $\| \cdot \|$ if for all conformable $X, Y$:*

$$\Pr\Big[ \|X S^\top S Y - X Y\| \leq \varepsilon \|X\| \|Y\| \Big] \geq 1 - \delta. \tag{5}$$

Oblivious sketches satisfying equation 5 are well-studied. In particular, the stable-rank-optimal results of Cohen et al. (2016) achieve equation 5 when factors have bounded stable rank. We deliberately keep the sketch dimension $m$ implicit until applying AMMM to clustering.

## 4.2 MAIN BOUND AND PROOF SKETCH

**Theorem 4.2** (AMMM from One-Step AMM). *Assume equation 5 holds for independent $S_1, \ldots, S_{k-1}$. Then with probability at least $1 - ck\delta$ for an absolute constant $c$:*

$$\big\|\widetilde{M}_k - M_k\big\| \leq \sum_{t=1}^{k-1} \varepsilon (1+\varepsilon)^{k-1-t} \|A_1 \cdots A_t\| \|A_{t+1} \cdots A_k\|. \tag{6}$$

*In particular, by submultiplicativity:*

$$\big\|\widetilde{M}_k - M_k\big\| \leq \big((1+\varepsilon)^{k-1} - 1\big) \prod_{i=1}^{k} \|A_i\|. \tag{7}$$

*Proof.* Define $M_t := A_1 \cdots A_t$ and $\widetilde{M}_t := A_1 S_1^\top S_1 \cdots A_{t-1} S_{t-1}^\top S_{t-1} A_t$. The proof proceeds by:

1. Adding and subtracting $M_t S_t^\top S_t A_{t+1}$ at each step

2. Applying the one-step guarantee equation 5 twice (once for $M_t$, once for the error $\widetilde{M}_t - M_t$)

3. Obtaining the recurrence:

$$\|\widetilde{M}_{t+1} - M_{t+1}\| \leq \varepsilon\big(\|M_t\| + \|\widetilde{M}_t - M_t\|\big) \|A_{t+1}\|.$$

4. Recursively expanding this recurrence yields equation 6

5. Applying submultiplicativity $\|M_t\| \leq \prod_{i=1}^{t} \|A_i\|$ gives equation 7

The probability bound follows from applying the one-step result $O(k)$ times with a union bound. $\square$

Theorem 4.2 establishes an upper bound for AMMM, but at this time a corresponding lower bound for general $k$ is an open problem. Notably for $k = 2$, Theorem 4.2's upper bound corresponds to the tight lower bound Cohen et al. (2016).

**Corollary 4.3** (Power Chains). *If $A_1 = \cdots = A_k = B$, then with the same probability as Theorem 4.2:*

$$\big\|B S_1^\top S_1 B S_2^\top S_2 \cdots B S_{k-1}^\top S_{k-1} B - B^k\big\| \leq \big((1+\varepsilon)^{k-1} - 1\big) \|B\|^k. \tag{8}$$

*Proof.* Direct application of Theorem 4.2 with all $A_i = B$. $\square$

The inequality equation 6 shows that *local* one-step AMM errors accumulate controllably rather than exploding across the chain. This enables replacing iterative multiplications (with dimension $n$) with streaming sketch accumulation followed by the multiplication of several small sketch matrices (where most products have dimension $m \ll n$). Moreover, the sketch components of AMMM can

be accumulated from a turnstile stream. Corollary 4.3 ensures accuracy preservation for downstream tasks like clustering.

To use AMMM in practice, instantiate the one-step hypothesis equation 5. For example, Cohen et al. (2016) show that for sketches with dimension determined by the stable rank of factors, equation 5 holds. Plugging such an AMM into Theorem 4.2 yields AMMM for the interleaved form equation 4. We explore sketch dimension choices for clustering in §5, where the pass-memory tradeoff becomes meaningful.

## 5 FROM STABLE RANK AMM TO STREAMING CLUSTERING

We combine the stable rank dependent approximate matrix multiplication result of Cohen et al. (2016) with the AMMM bounds from §4 to obtain explicit, sublinear embedding dimensions that preserve the row distance threshold in Assumption 3.2. Throughout, set $X^\star := B^r$.

### 5.1 STABLE RANK AMM

We use a subgaussian sketch bound that depends only on the operator norm and the stable rank. We begin by recalling the Oblivious Subspace Embedding (OSE) moment property.

**Definition 5.1** (OSE moment property). *A distribution $\mathcal{D}$ over $\mathbb{R}^{m \times n}$ has the $(\varepsilon, \delta, s, \ell)$ OSE moment property if for every matrix $U \in \mathbb{R}^{n \times d}$ with orthogonal columns,*

$$\mathbb{E}_{\Pi \sim \mathcal{D}} \|(\Pi U)^T (\Pi U) - I\|^\ell < \varepsilon^\ell \delta.$$

We now state the main lemma from Cohen et al. (2016), which will be used to obtain our embedding dimensions.

**Theorem 5.2** (Theorem 1 of Cohen et al. (2016)). *Let $k \geq 1$ and $\varepsilon, \delta \in (0, 1/2)$. Suppose $\Pi \in \mathbb{R}^{m \times n}$ is drawn from a distribution that satisfies the $(\varepsilon, \delta, 2k, \ell)$ OSE moment property for some $\ell \geq 2$. Then for any $A, B$,*

$$\Pr \left\{ \|(\Pi A)^T (\Pi B) - A^T B\|_{op} \leq \varepsilon \sqrt{\left( \|A\|_{op}^2 + \frac{\|A\|_F^2}{k} \right) \left( \|B\|_{op}^2 + \frac{\|B\|_F^2}{k} \right)} \right\} \geq 1 - \delta. \quad (9)$$

*Moreover, for a Rademacher or subgaussian sketch,*

$$m \geq C \frac{k + \log(1/\delta)}{\varepsilon^2} \quad \Longrightarrow \quad \Pi \text{ satisfies the } (\varepsilon, \delta, 2k, \Theta(k + \log(1/\delta))) \text{ OSE moment property,}$$
$$(10)$$

*which implies equation 9. If $k \geq \max\{\text{sr}(A), \text{sr}(B)\}$, the bound simplifies to*

$$\|(\Pi A)^T (\Pi B) - A^T B\|_{op} \leq 2\varepsilon \|A\|_{op} \|B\|_{op}. \quad (11)$$

We next derive a high probability bound on the stable rank of the shifted adjacency matrix $B$.

**Lemma 5.1** (High probability upper bound on $\text{sr}(B)$). *Let $G \sim \text{SBM}(n, K, \{s_\ell\}, p, q)$ with adjacency matrix $A$. Let $B, L, \text{and} R$ be defined as in equation 1. Write $s_* = \max_\ell s_\ell$ and define*

$$\sigma^2 = \max\{p(1-p), q(1-q)\}.$$

*There exist constants $C_1, C_2 > 0$ such that, with probability at least $1 - 3n^{-3}$,*

$$\text{sr}(B) = \frac{\|B\|_F^2}{\|B\|_{op}^2} \leq \frac{(p-q)^2 \sum_{\ell=1}^K s_\ell^2 + n^2 \sigma^2 + C_1 n \sqrt{\log n}}{\left((p-q)s_* - C_2 \sigma \sqrt{n}\right)^2}. \quad (12)$$

*In particular, under the signal-dominated regime $(p-q)s_* \gg \sigma\sqrt{n}$:*

- **Balanced case** *($s_\ell = n/K$):*

$$\text{sr}(B) = O\left(K + \frac{K^2 \sigma^2}{(p-q)^2}\right).$$

*If additionally $(p-q)^2 \gtrsim K\sigma^2$, then $\text{sr}(B) = \Theta(K)$. More generally, under the weaker detectability condition $(p-q)^2 \gtrsim K^2\sigma^2/n$ (which follows from $(p-q)s_* \gg \sigma\sqrt{n}$ with $s_* = n/K$), we have $\text{sr}(B) = o(n)$, confirming sublinearity.*

- **Highly imbalanced case** $(\sum_\ell s_\ell^2 \asymp s_*^2)$:

$$\mathrm{sr}(B) = O\left(1 + \frac{n^2\sigma^2}{(p-q)^2 s_*^2}\right).$$

If $(p-q)s_* \gg \sigma n$, then $\mathrm{sr}(B) = \Theta(1)$.

*Proof.* Deferred to Appendix C. $\qquad\square$

The next lemma converts an operator norm bound on a product approximation into a bound on all pairwise row distances. This lets us transfer a global AMMM error bound into a guarantee for the separation scale.

**Lemma 5.2** (Product implies row distances)**.** *For $M, \widetilde{M} \in \mathbb{R}^{n \times d}$ with $E = \|\widetilde{M} - M\|_{op}$, for all $i \neq j$,*

$$\left| \|\widetilde{M}_{i,\cdot} - \widetilde{M}_{j,\cdot}\|_2 - \|M_{i,\cdot} - M_{j,\cdot}\|_2 \right| \leq \sqrt{2}\, E. \tag{13}$$

*Proof.* Let $u = e_i - e_j$. Then

$$\left| \|u^T \widetilde{M}\|_2 - \|u^T M\|_2 \right| \leq \|u^T(\widetilde{M} - M)\|_2 \leq \|u\|_2\, E = \sqrt{2} E.$$

$\qquad\square$

## 5.2 MAIN STREAMING THEOREM (EXPLICIT SUBLINEAR $m$)

We now state the main streaming result with an explicit embedding dimension.

**Theorem 5.3** (Separation scale preservation for streaming Algorithm B; explicit $m$)**.** *Assume Assumption 3.2 holds for $X^\star = B^r$ with parameters $(a, b, \Delta_r)$ and failure probability $\eta$. Fix $\varepsilon \in (0, 1/4)$ and $\delta \in (0,1)$. Suppose $S_1, \ldots, S_{r-1} \in \mathbb{R}^{m \times n}$ are independent Rademacher matrices with*

$$m \geq C\, \frac{\mathrm{sr}(B) + \log((r-1)/\delta)}{\varepsilon^2}. \tag{14}$$

*Then, with probability at least $1 - \eta - 2\delta$, for all $i \neq j$,*

$$\left| \|\widetilde{B}_{i,\cdot}^{(r)} - \widetilde{B}_{j,\cdot}^{(r)}\|_2 - \|B_{i,\cdot}^r - B_{j,\cdot}^r\|_2 \right| \leq \sqrt{2}\big((1+2\varepsilon)^{r-1} - 1\big)\|B\|_{op}^r. \tag{15}$$

*If, in addition, there exist $0 < a < b$ such that*

$$\sqrt{2}\big((1+2\varepsilon)^{r-1} - 1\big)\|B\|_{op}^r \leq \frac{b-a}{4}\Delta_r, \tag{16}$$

*then the threshold $\Delta_r$ still separates within-cluster and across-cluster pairs for $\widetilde{B}^{(r)}$, and single linkage at cut $\Delta_r$ recovers the largest cluster.*

*Proof.* Deferred to Appendix C. $\qquad\square$

Note that Theorem 4.2 states the bound on embedding dimension $m$ in terms of dense Rademacher $\{\pm 1\}$ matrices. A similar argument yields a slightly larger bound for sparse Johnson-Lindenstrauss transforms, such as those implemented with CountSketch Cohen et al. (2016). While such transforms will yield better runtime performance in practice, we have omitted it for space.

## 5.3 $r$-PASS VARIANT WITH EXPLICIT $m$

We now present a similar separation guarantee for Algorithm C. This variant is practical when $A$ is static and its edge stream can be warehoused and read in multiple passes, and the cost of each sparse product of the form $AS^T$ is low. In this setting, sketching $B^r S^T$ leads to a lower effective bound on $m$ at the cost of requiring $r$ passes over $A$.

**Theorem 5.4** (Separation scale preservation for $r$-pass Algorithm C; explicit $m$)**.** *Let $X^\star = B^r \in \mathbb{R}^{n \times n}$ with rows $x_i^\star$. Assume Assumption 3.2 holds for $X^\star$ with parameters $(a, b, \Delta_r)$ and success probability $1 - \eta$. In other words, with probability at least $1 - \eta$,*

$$\|x_i^\star - x_j^\star\|_2 \leq a\, \Delta_r \quad \text{(within)}, \qquad \|x_i^\star - x_j^\star\|_2 \geq b\, \Delta_r \quad \text{(across)}.$$

*Let $S \in \mathbb{R}^{m \times n}$ have independent mean-zero variance-$1/m$ subgaussian entries with subgaussian norm at most $\kappa$. Fix $\varepsilon \in (0, 1)$ and $\delta \in (0, 1)$. If*

$$m \geq \frac{C(\kappa)}{\varepsilon^2} \big( \log n + \log(1/\delta) \big), \tag{17}$$

*then with probability at least $1 - \eta - \delta$, for all $i \neq j$,*

$$(1 - \varepsilon)\|x_i^\star - x_j^\star\|_2 \leq \|S(x_i^\star - x_j^\star)\|_2 \leq (1 + \varepsilon)\|x_i^\star - x_j^\star\|_2.$$

*If, in addition,*

$$(1 + \varepsilon)a < (1 - \varepsilon)b, \tag{18}$$

*then there exists a threshold*

$$\widetilde{\Delta} \in \big( (1 + \varepsilon)a\, \Delta_r, \; (1 - \varepsilon)b\, \Delta_r \big)$$

*that separates within-cluster and across-cluster pairs for the sketched rows $\{Sx_i^\star\}$, and Algorithm C recovers the largest cluster.*

*Proof.* Deferred to Appendix C. $\qquad\square$

**Takeaway.** The explicit bounds equation 14, equation 17 coupled with the fact that $\mathrm{sr}(B) \leq rank(B) = \mathrm{o}(n)$ show that the embeddings are sublinear: $m = \tilde{O}(\mathrm{sr}(B)/\varepsilon^2)$ for the streaming scheme and $m = O(\varepsilon^{-2} \log n)$ for the $r$-pass scheme. Both preserve the decision threshold $\Delta_r$ used by the unsketched method.

## 6 EXPERIMENTAL EVALUATION

We empirically validate that (i) the streaming and $r$-pass variants preserve the separation scale from Assumption 3.2, (ii) the required embedding dimension $m$ is *sublinear* (scales with $\mathrm{sr}(B)$ or $\log n$), and (iii) the resulting clustering recovers the largest community efficiently.

### 6.1 METHODOLOGY

**Synthetic data generation.** We generate graphs from $\mathrm{SBM}(n, K, \{s_\ell\}, p, q)$ as in §2, considering the following parameters. Our SBM generator does not sample self loops, although our theory allows for them. We fix $n = 10^4$ and choose $K \in \{5, 10, 20\}$. We vary $s_*/s_{\min} \in \{1, 2, 4, 8\}$. In our SBM generator, we set an imbalance parameter $\mathrm{imbalance} \in [0, 1]$ interpolating community sizes: $\mathrm{imbalance} = 0$ gives equal sizes, while $\mathrm{imbalance} = 1$ yields a largest community of size $s^*$ and others of size at least $(q/p)\, s^*$. Choose $p = q + C_1$ where $C_1 > 0.1$ to sweep the operator SNR $\frac{(p-q)s_*}{\sqrt{p(1-q)}\sqrt{n}}$ from below to above the recovery threshold. We consider the matrix power $r \in \{2, 3, 4\}$. For each setting we run 5 independent trials with different random seeds and report the average of the evaluation metrics.

**Methods compared.** We implemented Algorithms B and C with independent Rademacher sketches. We size $m$ for both using the *theory line* from Theorem 5.3 and Theorem 5.4, respectively:

$$m_{\mathrm{inter}} = C\, \frac{\widehat{\mathrm{sr}}(B) + \log\big((r-1)/\delta\big)}{\varepsilon^2}, \qquad m_{\mathrm{end}} = \frac{8}{\varepsilon^2}\Big( \log \tbinom{n}{2} + \log(2/\delta) \Big).$$

where $C$ is some constant and $\widehat{\mathrm{sr}}(B) = \|B\|_{\mathrm{F}}^2 / \|\widehat{B}\|_{\mathrm{op}}^2$ with $\|\widehat{B}\|_{\mathrm{op}}$ estimated by power iteration (10 steps sufficed in all runs). We also sweep $m \in \{0.5, 1, 2\} \cdot m_*$ for $m_* \in \{m_{\mathrm{inter}}, m_{\mathrm{end}}\}$. In all experiments we fixed $\varepsilon \in \{0.1, 0.2, 0.3\}$ and $\delta = 0.05$ unless otherwise noted.

**Metrics.** We collect the largest cluster output by our algorithms as $\widehat{C}_*$ and compare it with the true largest community $C_*$. We report (i) *Largest–cluster recall* $\text{Rec}_* = \frac{|\widehat{C}_* \cap C_*|}{|C_*|}$; (ii) *Largest–cluster precision* $\text{Prec}_* = \frac{|\widehat{C}_* \cap C_*|}{|\widehat{C}_*|}$; (iii) *Largest-cluster F1*, $\text{F1}_* = 2\frac{\text{Rec}_* \text{Prec}_*}{\text{Rec}_* + \text{Prec}_*}$; (iv) and *Row–gap margin* $M = \min_{i \not\sim j}\text{dist}(i,j) - \max_{i \sim j}\text{dist}(i,j)$ at the chosen $\widehat{\Delta}$.

## 6.2 Results

**Largest cluster recovery.** Figure 1 plots largest-cluster recall, precision, and F1 for fixed $(n, K, p, q, r)$ as we sweep $m$ for both Algorithms B and C. Figure 1c shows dimension normalization across $n$ and verifies that recall saturates near 1 once $m \gtrsim c\,\widehat{\text{sr}}(B)/\varepsilon^2$. End–sketch acts as a comparator with $m = O(\varepsilon^{-2}\log n)$.

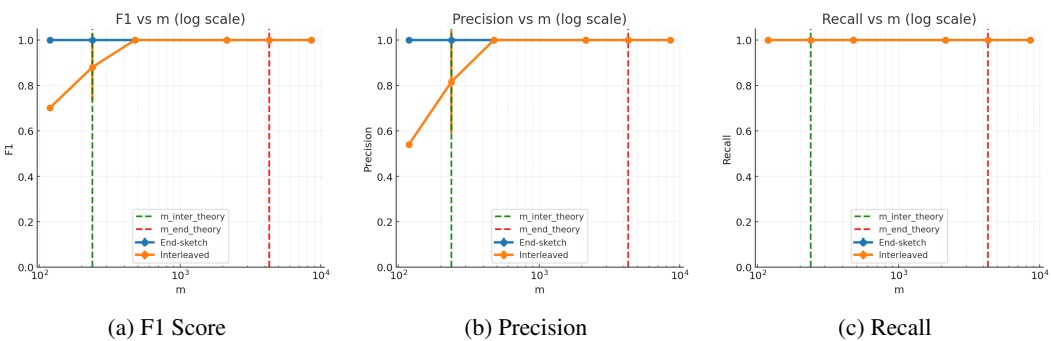

(a) F1 Score        (b) Precision        (c) Recall

Figure 1: Parameters: $n = 10^4, K = 5, \text{imbalance} = 0.5, p = 0.3, q = 0.01$

Figure 1 shows that the $r$-pass pipeline always perfectly recovers the largest cluster as long as the embedding dimension bound is satisfied. Moreover, it shows that the performance of the streaming approach improves drastically as the embedding dimension increases. This confirms the noise introduced by interleaved sketches is inversely proportional to the embedding dimension.

**The effects of imbalance and power.** Figure 2 shows the effects of varying the imbalance, which is directly proportional to $s_*/s_{\min}$, and how it interacts with the matrix power $r$. We observe that as the imbalance increases, the largest community grows, so the threshold $\Delta = \frac{1}{2}(p-q)^r(s^*)^{r-\frac{1}{2}}$ rises and points from that community become relatively closer to one another than to the rest; this makes them easier to separate, so the $F1$ score improves. At the same time the graph is dominated by that one community, so the matrix $B$ is effectively simpler and its stable rank drops (most variation now comes from one direction). Consequently, higher imbalance yields both better clustering and a smaller sketch size when we choose $m \propto \text{sr}(B)$. We sweep $r \in \{2, 3, 4\}$. Larger $r$ increases separation scale $\Delta_r$ (Lemma 3.1) but also increases sensitivity to operator norm growth in the AMMM bound. We show recall and margin $M$ vs. $r$. The value of $m$ is dictated by the formula $m_{\text{inter}}$.

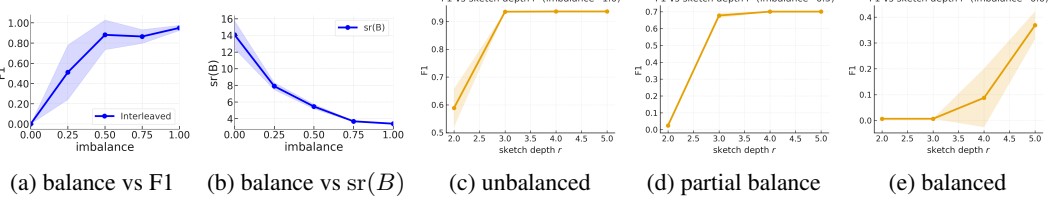

(a) balance vs F1    (b) balance vs $\text{sr}(B)$    (c) unbalanced    (d) partial balance    (e) balanced

Figure 2: 2a and 2b show the relationship between imbalance and F1 and $\text{sr}(B)$, respectively. 2c, 2d, and 2e show how F1 improves with $r$ for balanced vs. unbalanced graphs.

**Separation scale preservation.** Figure 3 demonstrates that given a reasonable embedding dimension the same threshold we used for clustering the graph still applied to both Algorithms B and C. This allows us to visualize Theorem 5.3 and 5.4 at the level of the separation scale.

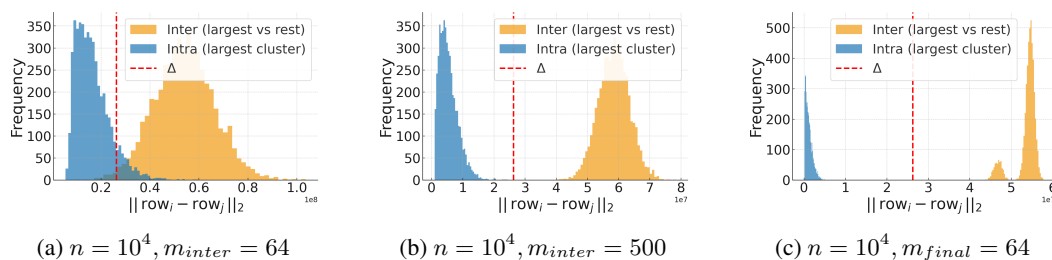

(a) $n = 10^4, m_{inter} = 64$      (b) $n = 10^4, m_{inter} = 500$      (c) $n = 10^4, m_{final} = 64$

Figure 3: Row distance histograms for different embedding dimensions and embedding approaches.

Across all experiments, the intra-cluster distances in the largest cluster concentrate well to the left of the fixed separation scale $\Delta_r$, while the inter-cluster distances lie to the right when the sketch is sufficiently large, giving clear separation. When we shrink the dimension to $64$, the $r$-pass variant still shows a clean gap, but the streaming algorithm develops visible overlap around $\Delta$, indicating that too small an $m$ injects enough noise to blur the boundary. Thus, in this setting the $\Delta_r$ separation scale is preserved for both methods, although the streaming case requires larger $m$, as suggested by theory.

## 7 CONCLUSION

We have analyzed and demonstrated a novel turnstile streaming power iteration clustering algorithm utilitizing approximate multi-matrix multiplication and oblivious subspace embeddings. Our experiments directly demonstrate a practical sublinear embedding dimension for largest cluster recovery, validating Theorems 5.3 and 5.4. Obvious future work includes an analysis of the recovery of all clusters, as well as an examination of the removal of the awkward centering step required in Algorithm 1. It would also be desirable to find a space lower bound for the approximate multi matrix multiplication problem. Furthermore, although our experiments are relatively unoptimized and sequential, Algorithms B and C are easy to implement in a distributed computing model and will be interesting to demonstrate on billion-scale data.

## 8 ACKNOWLEDGEMENTS

This work was performed under the auspices of the U.S. Department of Energy by Lawrence Livermore National Laboratory under Contract DE-AC52-07NA27344 (LLNL-CONF-2011640), and was supported by LLNL LDRD project 24-ERD-024.

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

## A    RELATED WORK

For a symmetric, square graph matrix $M$, power iteration amounts to estimating the eigenvectors by repeated multiplication by a oblivious subspace embedding matrix $S$ so that the columns of $M^r S$ for some small $r$ can stand in for $M$'s top eigenvectors. Many investigators have analyzed orthonormalizing and/or decomposing $M^r S$ to prove approximation guarantees to the eigenvectors (Halko et al., 2011; Woodruff et al., 2014), while others focus on various guarantees in the context of k-means clustering (Boutsidis et al., 2015; Macgregor, 2023; Lin & Cohen, 2010). While the majority of these analyses require dense i.i.d. Guassian transforms $S$, realizing and applying such a matrix is impractical for very large graphs. Fortunately, dense Gaussian matrices are but one distribution satisfying the conditions of the famous Johnson-Lindenstrauss lemma (Johnson et al., 1984). Achlioptas (2003) introduced scaled dense $\{+1, -1\}$ matrices of zero-mean Rademacher variables, while several works introduced sparsity and eventually worked out tight sparsity bounds (Li et al., 2006; Nelson & Nguyễn, 2013; Kane & Nelson, 2014). In parallel, Clarkson & Woodruff (2017) introduced a maximally sparse matrix distribution based upon the famous Count-Sketch data structure (Charikar et al., 2002), which can be shown to satisfy the same sparsity bounds Cohen et al. (2016). Some investigators have focused on empirically demonstrating the performance of power iteration utilizing these more practical sparse Johnson-Lindenstrauss transforms (Zhang et al., 2018) or polynomials thereof (Chen et al., 2019), although these analyses are lacking analytic guarantees for clustering or other downstream tasks.

## B  STREAMING UPDATES

Here we make the streaming nature of Subroutine B more explicit. The approximate power iteration subroutine's goal is to output the product $\widetilde{B}^{(r)} \equiv BS_1^\top \prod_{i=1}^{r-1} S_i BS_{i+1}^\top$. First, we recall that $B \leftarrow A - q\mathbf{1}\mathbf{1}^\top$. As the $q\mathbf{1}\mathbf{1}^\top$ component is constant and all operations are linear, we can safely defer its incorporation to the sketches to the end and primarily reason about $A$. Here Subroutine D shows how Subroutine B handles a stream $\sigma$ of updates of the form $(i, j, w)$ to $A$. Let $\widehat{A}$ be a variable representing the matrix formed from reading updates from $\sigma$, which could be received in any order. Upon reading all updates in $\sigma$, $\widehat{A} = A$. In each such update, $\widehat{A}_{i,j}$ receives the modification $w \pm 1$. As $A$ is the adjacency matrix of an unweighted SBM, we assert without a loss of generality that an update $(i, j, w)$ implies $(j, i, w)$. Furthermore, we state that positive updates $(i, j, 1)$ are only received when $\widehat{A}_{i,j} = 0$ and negative updates $(i, j, -1)$ are only received when $\widehat{A}_{i,j} = 1$. This is tantamount to asserting that $\sigma$ represents a stream of edge additions and deletions, which can only occur if the given edge does not/does exist, respectively. Although the method remains valid without this last assumption, without it $\widehat{A}$ could at some point represent a non-simple graph.

Given these assumptions, Subroutine D updates each of the temporary sketch variables $\widehat{A}^{(1)} \equiv \widehat{A}S_1$ and $\widehat{A}^{(i)} \equiv S_{i-1}^\top \widehat{A}S_i^\top$ as it receives updates from $\sigma$. We represent each of these updates $(i, j, w)$ as the rank one matrix $U \leftarrow w\mathbf{e}_i\mathbf{e}_j^\top$, where $\mathbf{e}_i$ and $\mathbf{e}_j$ refer to the usual basis vectors. Applying the appropriate sketch transforms to this matrix provides us with additive updates to $\widehat{A}^{(1)}, \ldots, \widehat{A}^{(r-1)}$. After receiving all updates, we incorporate the constant shift $q\mathbf{1}\mathbf{1}^\top$ and return the product $BS_1^\top \prod_{i=1}^{r-1} S_i BS_{i+1}^\top$.

---

**Subroutine D** STREAMING UPDATES TO POWER ITERATION

1: **Input:** Integral $r > 1$, $m = O(\mathrm{sr}(B)\varepsilon^{-2})$, stream $\sigma$ of updates to $A$
2: Sample $S_1, \ldots, S_{r-1} \in \mathbb{R}^{m \times n}$
3: $\widehat{A}^{(1)} \leftarrow 0^{n \times m}$
4: **for** $i \in \{2, \ldots, r-1\}$ **do**
5: $\quad \widehat{A}^{(i)} \leftarrow 0^{m \times m}$
6: **end for**
7: **for** update $(i, j, w) \in \sigma$ **do**
8: $\quad U \leftarrow w\mathbf{e}_i\mathbf{e}_j^\top$
9: $\quad \widehat{A}^{(1)} \leftarrow US_1^\top$
10: $\quad$ **for** $i \in \{2, \ldots, r-1\}$ **do**
11: $\quad\quad \widehat{A}^{(i)} \leftarrow S_{i-1}^\top US_i^\top$
12: $\quad$ **end for**
13: **end for**
14: $Q \leftarrow -q\mathbf{1}\mathbf{1}^\top$
15: $BS_1^\top \leftarrow \widehat{A}^{(1)} + QS_1^\top$
16: **for** $i \in \{1, \ldots, r-2\}$ **do**
17: $\quad S_i BS_{i+1}^\top \leftarrow \widehat{A}^{(i)} + S_i QS_{i+1}^\top$
18: **end for**
19: $\widetilde{B}^{(r)} \leftarrow BS_1^\top \prod_{i=1}^{r-1} S_i BS_{i+1}^\top$
20: Output $\widetilde{B}^{(r)}$

---

## C  DEFERRED PROOFS

Herein we present the proofs of major results excluded from the main body due to space constraints.

**Lemma 3.1 Norms of $L$ and row separation.**  We have $\|L\|_{\mathrm{op}} = (p-q)s_*$ and $\|L\|_\mathrm{F}^2 = (p-q)^2 \sum_{\ell=1}^K s_\ell^2$. Moreover, rows of $L^r$ are constant within each cluster and, if $i \in C_\ell$ and $j \in C_{\ell'}$ with $\ell \neq \ell'$,

$$\|L_{i,\cdot}^r - L_{j,\cdot}^r\|_2 = (p-q)^r \sqrt{s_\ell^{2r-1} + s_{\ell'}^{2r-1}}. \tag{19}$$

*Proof.* We know by definition that:

$$L_{ij} = \begin{cases} (p-q) & \text{if } i \sim j \\ 0 & \text{if } i \not\sim j \end{cases}$$

It follows that $L$ is a block diagonal matrix. We can represent it as follows:

$$L = \begin{bmatrix} (p-q)\mathbf{1}_{s_1 \times s_1} & \mathbf{0} & \cdots & \mathbf{0} \\ \mathbf{0} & (p-q)\mathbf{1}_{s_2 \times s_2} & \ddots & \vdots \\ \vdots & \ddots & \ddots & \mathbf{0} \\ \mathbf{0} & \cdots & \mathbf{0} & (p-q)\mathbf{1}_{s_K \times s_K} \end{bmatrix} = \begin{bmatrix} u_1 v_1^T & 0 & \cdots & 0 \\ 0 & u_1 v_1^T & \ddots & \vdots \\ \vdots & \ddots & \ddots & 0 \\ 0 & \cdots & 0 & u_K v_K^T \end{bmatrix}$$

where

$$u_j = \begin{bmatrix} 1 \\ \vdots \\ 1 \end{bmatrix}, \quad v_j = \begin{bmatrix} (p-q) \\ \vdots \\ (p-q) \end{bmatrix} \in \mathbb{R}^{s_l}.$$

The operator norm is then the largest of the singular values of the block matrices. Since each block matrix in $L$ is rank 1 then the operator norm bound follows by noting that

$$\|u_j v_j^T\|_{op} = \|u_j\|_2 \|v_j\|_2 = \sqrt{s_j}(p-q)\sqrt{s_j} = (p-q)s_j$$

Furthermore, from the structure of $L$ we can deduce that

$$\|L\|_F^2 = \sum_{(i,j)} L_{ij}^2 = \sum_{\ell=1}^K (p-q)^2 s_\ell^2$$

The second part of the result follows from a simple counting argument after explicitely writing the entries of $L_{ij}^r$ $\qquad\square$

**Corollary 4.3 Power chains.** If $A_1 = \cdots = A_k = B$, then with the same probability as Theorem 4.2,

$$\left\| B S_1^\top S_1 B S_2^\top S_2 \cdots B S_{k-1}^\top S_{k-1} B - B^k \right\| \leq \left( (1+\varepsilon)^{k-1} - 1 \right) \|B\|^k. \tag{20}$$

*Proof.* Let $\widetilde{M}_1 := B$, $\widetilde{M}_{t+1} := \widetilde{M}_t S_t^\top S_t B$, and $M_t := B^t$. With $E_t := \|\widetilde{M}_t - M_t\|$ and $E_1 = 0$,

$$\widetilde{M}_{t+1} - M_{t+1} = (\widetilde{M}_t - M_t)S_t^\top S_t B + \underbrace{M_t S_t^\top S_t B - M_t B}_{\text{one–step error}}.$$

Condition on $(\widetilde{M}_t, M_t)$ (independent of $S_t$) and apply equation 4.1 twice. By a standard union bound argument we have that with prob. $\geq 1 - 2\delta$,

$$E_{t+1} \leq \varepsilon \|B\| E_t + \varepsilon \|B\|^{t+1}.$$

After recursively expanding the above relation satisfied by the $E_t$ combined with the submultiplicativity of the norm we have:

$$E_k \leq ((1+\varepsilon)^{k-1} - 1)\|B\|^k.$$

The probability bound follows in a similar fashion as in Theorem 4.2 $\qquad\square$

**Lemma 5.1 High–probability upper bound on** $\mathrm{sr}(B)$**.** Let $G \sim \mathrm{SBM}(n, K, \{s_\ell\}, p, q)$ with adjacency matrix $A$. Let $B$, $L$, and $R$ be defined as in equation 1. Write $s_* = \max_\ell s_\ell$ and define

$$\sigma^2 = \max\{p(1-p), q(1-q)\}.$$

There exist constants $C_1, C_2 > 0$ such that, with probability at least $1 - 3n^{-3}$,

$$\mathrm{sr}(B) = \frac{\|B\|_F^2}{\|B\|_{op}^2} \leq \frac{(p-q)^2 \sum_{\ell=1}^K s_\ell^2 + n^2 \sigma^2 + C_1 n\sqrt{\log n}}{\left((p-q)s_* - C_2 \sigma\sqrt{n}\right)^2}. \tag{21}$$

In particular, under the signal-dominated regime $(p-q)s_* \gg \sigma\sqrt{n}$:

- **Balanced case** ($s_\ell = n/K$):

$$\mathrm{sr}(B) = O\left(K + \frac{K^2\sigma^2}{(p-q)^2}\right).$$

If additionally $(p-q)^2 \gtrsim K\sigma^2$, then $\mathrm{sr}(B) = \Theta(K)$. More generally, under the weaker detectability condition $(p-q)^2 \gtrsim K^2\sigma^2/n$ (which follows from $(p-q)s_* \gg \sigma\sqrt{n}$ with $s_* = n/K$), we have $\mathrm{sr}(B) = o(n)$, confirming sublinearity.

- **Highly imbalanced case** ($\sum_\ell s_\ell^2 \asymp s_*^2$):

$$\mathrm{sr}(B) = O\left(1 + \frac{n^2\sigma^2}{(p-q)^2 s_*^2}\right).$$

If $(p-q)s_* \gg \sigma n$, then $\mathrm{sr}(B) = \Theta(1)$.

*Proof.* We first lower bound $\|B\|_{\mathrm{op}}$ and then upper bound $\|B\|_F^2$.

**Lower bound on $\|B\|_{\mathrm{op}}$:** By Weyl's inequality,

$$\lambda_1(B) \geq \lambda_1(L) - \|R\|_{\mathrm{op}}.$$

For the expectation $L$ of the centered matrix $B$, one has $\lambda_1(L) = \|L\|_{\mathrm{op}} = (p-q)s_*$. Moreover, for the noise part $R$,

$$\|R\|_{\mathrm{op}} \leq C_2\,\sigma\sqrt{n}$$

with probability at least $1 - n^{-3}$ (Lemma 2.1 of Mukherjee & Zhang (2024)). Thus, with the same probability, we have

$$\|B\|_{\mathrm{op}} = \lambda_1(B) \geq (p-q)s_* - C_2\,\sigma\sqrt{n}.$$

**Upper bound on $\|B\|_F^2$:** We start with the expansion that takes into account the symmetry of both $L$ and $R$:

$$\|B\|_F^2 = \|L\|_F^2 + \|R\|_F^2 + 2\langle L, R\rangle.$$

From the definition of $L$ it follows that:

$$\|L\|_F^2 = (p-q)^2 \sum_{\ell=1}^K s_\ell^2.$$

For $R$, whose entries are independent, mean zero, bounded, with $\mathrm{Var}(R_{ij}) \leq \sigma^2$, we have

$$\mathbb{E}\|R\|_F^2 = \sum_{i,j} \mathbb{E}[R_{ij}^2] = \sum_{i,j} \mathrm{Var}(R_{ij}) \leq n^2\sigma^2.$$

A Bernstein inequality for sums of independent bounded variables yields, with probability at least $1 - n^{-4}$,

$$\|R\|_F^2 \leq n^2\sigma^2 + C\,n\sqrt{\log n}.$$

Likewise, since $\langle L, R\rangle = \sum_{i,j} L_{ij}R_{ij}$ is a mean-zero sum of independent bounded variables supported on the block structure of $L$, Bernstein implies that, with probability at least $1 - n^{-4}$,

$$|\langle L, R\rangle| \leq C\,n\sqrt{\log n}.$$

Combining the bounds on the terms involved in the expansion yields:

$$\|B\|_F^2 \leq (p-q)^2 \sum_{\ell=1}^K s_\ell^2 + n^2\sigma^2 + C_1\,n\sqrt{\log n}.$$

Combining the above estimates using a union bound gives the final upper bound on $\mathrm{sr}(B)$ with failure probability at most $3n^{-3}$.

**Balanced case analysis:** If $s_\ell = n/K$ for all $\ell$, then $\sum_{\ell=1}^K s_\ell^2 = K(n/K)^2 = n^2/K$ and $s_* = n/K$. The numerator becomes:

$$(p-q)^2\frac{n^2}{K} + n^2\sigma^2 + O(n\sqrt{\log n}) = n^2\left(\frac{(p-q)^2}{K} + \sigma^2\right) + O(n\sqrt{\log n}).$$

In the signal-dominated regime $(p-q)s_* \gg \sigma\sqrt{n}$, i.e., $(p-q)n/K \gg \sigma\sqrt{n}$, the denominator is:

$$\left((p-q)s_* - C_2\sigma\sqrt{n}\right)^2 \asymp (p-q)^2 \frac{n^2}{K^2}.$$

Therefore:

$$\mathrm{sr}(B) \lesssim \frac{n^2\left(\frac{(p-q)^2}{K} + \sigma^2\right)}{(p-q)^2\frac{n^2}{K^2}} = K + \frac{K^2\sigma^2}{(p-q)^2}.$$

For $\mathrm{sr}(B) = \Theta(K)$, we require the second term to be $O(K)$, which holds when $(p-q)^2 \gtrsim K\sigma^2$. More generally, under the weaker condition $(p-q)^2 \gtrsim K^2\sigma^2/n$, we obtain $\mathrm{sr}(B) = O(K^2\sigma^2/(p-q)^2) = O(n)$. Since $(p-q)s_* \gg \sigma\sqrt{n}$ with $s_* = n/K$ gives $(p-q)^2n/K^2 \gg \sigma^2$, i.e., $(p-q)^2 \gg K^2\sigma^2/n$, we have $\mathrm{sr}(B) = o(n)$, confirming sublinearity.

**Highly imbalanced case:** When $\sum_\ell s_\ell^2 \asymp s_*^2$, the numerator is dominated by $(p-q)^2s_*^2 + n^2\sigma^2$, and the denominator (in the signal-dominated regime) is $(p-q)^2s_*^2$. Thus:

$$\mathrm{sr}(B) \lesssim \frac{(p-q)^2s_*^2 + n^2\sigma^2}{(p-q)^2s_*^2} = 1 + \frac{n^2\sigma^2}{(p-q)^2s_*^2}.$$

If $(p-q)s_* \gg \sigma n$, then the second term is $o(1)$, giving $\mathrm{sr}(B) = \Theta(1)$. $\qquad\square$

**Theorem 5.3 Separation scale preservation for Algorithm B; explicit $m$.** Assume Assumption 3.2 holds for $X^\star = B^r$ with parameters $(a, b, \Delta_r)$ and failure probability $\eta$. Fix $\varepsilon \in (0, 1/4)$ and $\delta \in (0, 1)$. Suppose $S_1, \ldots, S_{r-1}$ are independent Rademacher with

$$m \geq C\,\frac{\mathrm{sr}(B) + \log\left((r-1)/\delta\right)}{\varepsilon^2}. \tag{22}$$

Then, with probability at least $1 - \eta - 2\delta$, simultaneously for all $i \neq j$,

$$\left| \|\widetilde{B}_{i,\cdot}^{(r)} - \widetilde{B}_{j,\cdot}^{(r)}\|_2 - \|B_{i,\cdot}^r - B_{j,\cdot}^r\|_2 \right| \leq \sqrt{2}\left((1+2\varepsilon)^{r-1} - 1\right)\|B\|_{op}^r. \tag{23}$$

If, in addition, there exist some $0 < a < b$ such that

$$\sqrt{2}\left((1+2\varepsilon)^{r-1} - 1\right)\|B\|_{op}^r \leq \frac{b-a}{4}\Delta_r, \tag{24}$$

then the same threshold $\Delta_r$ separates within/across–cluster pairs for $\widetilde{B}^{(r)}$, and single–linkage at cut $\Delta_r$ recovers the largest cluster.

*Proof.* First observe that for any integer $t$:

$$\mathrm{sr}(B^t) = \frac{\|B^t\|_F^2}{\|B^t\|_{op}^2} \leq \frac{\left(\|B^{t-1}\|_{op}\|B\|_F\right)^2}{\|B\|_{op}^{2t}} \leq \mathrm{sr}(B).$$

Now choosing $m$ according to equation 14 in Theorem 5.2, forces each $S_t$ to satisfy equation 11 with accuracy $2\varepsilon$ and failure $\delta/(r-1)$. Corollary 4.3 then gives $\|\widetilde{B}^{(r)} - B^r\| \leq ((1+2\varepsilon)^{r-1} - 1)\|B\|^r$ with prob. $\geq 1 - 2\delta$. Apply Lemma 5.2 to obtain equation 23. Assuming there are some $0 < a < b$ satisfying equation 24, then every within–cluster distance remains $\leq \Delta_r$ and every across–cluster distance remains $\geq \Delta_r$. Union–bound the two events. $\qquad\square$

**Theorem 5.4** Let $X^\star = B^r \in \mathbb{R}^{n\times n}$ with rows $x_i^\star$. Assuming the row gap assumption 3.2 holds for $X^\star$ with parameters $(a, b, \Delta_r)$ and success probability $1 - \eta$, that is: with probability $\geq 1 - \eta$,

$$\|x_i^\star - x_j^\star\|_2 \leq a\,\Delta_r \quad \text{(within)}, \qquad \|x_i^\star - x_j^\star\|_2 \geq b\,\Delta_r \quad \text{(across)},$$

for some $0 < a < 1 < b$. Let $S \in \mathbb{R}^{m\times n}$ have i.i.d. mean-zero, variance-$1/m$ subgaussian entries with subgaussian norm at most $\kappa$ (a fixed constant). Fix $\varepsilon \in (0, 1)$ and $\delta \in (0, 1)$. If

$$m \geq \frac{C(\kappa)}{\varepsilon^2}\left(\log n + \log\frac{1}{\delta}\right), \tag{25}$$

then with probability at least $1 - \eta - \delta$ the following holds simultaneously for all $i \neq j$:

$$(1 - \varepsilon)\, \|x_i^\star - x_j^\star\|_2 \ \leq \ \|S(x_i^\star - x_j^\star)\|_2 \ \leq \ (1 + \varepsilon)\, \|x_i^\star - x_j^\star\|_2.$$

Consequently, if

$$(1 + \varepsilon)\, a \ < \ (1 - \varepsilon)\, b, \tag{26}$$

There exists a threshold $\widetilde{\Delta} \in \big((1 + \varepsilon)a\,\Delta_r,\ (1 - \varepsilon)b\,\Delta_r\big)$ that separates within- and across-cluster pairs for the sketched rows $\{Sx_i^\star\}$; hence, the thresholding step in Algorithm 1 recovers the largest cluster.

*Proof.* **Step 1: JL preservation for a fixed finite set.** Let $\mathcal{J}$ be the event that $S$ preserves all pairwise distances among $\{x_i^\star\}_{i=1}^n$ within a factor $1 \pm \varepsilon$. For subgaussian maps with parameter $\kappa$, the Johnson-Lindenstrauss lemma (via Hanson-Wright or standard subgaussian concentration) ensures that equation 25 implies

$$\mathbb{P}(\mathcal{J}) \ \geq \ 1 - \delta, \quad \text{and on } \mathcal{J}: \quad (1 - \varepsilon)\|u\| \leq \|Su\| \leq (1 + \varepsilon)\|u\| \ \ \forall u \in \{x_i^\star - x_j^\star\}.$$

**Step 2: Intersect with Assumption 3.2** Let $\mathcal{A}$ be the Assumption3.2 event (success $\geq 1 - \eta$). A union bound yields

$$\mathbb{P}(\mathcal{A} \cap \mathcal{J}) \ \geq \ 1 - \eta - \delta.$$

We henceforth work on $\mathcal{A} \cap \mathcal{J}$.

**Step 3: Threshold separation after sketching.** For any within-cluster pair, $\mathcal{A}$ gives $\|x_i^\star - x_j^\star\|_2 \leq a\,\Delta_r$, and then $\mathcal{J}$ implies

$$\|S(x_i^\star - x_j^\star)\|_2 \ \leq \ (1 + \varepsilon)a\,\Delta_r.$$

For any across-cluster pair, $\mathcal{A}$ gives $\|x_i^\star - x_j^\star\|_2 \geq b\,\Delta_r$, and then $\mathcal{J}$ implies

$$\|S(x_i^\star - x_j^\star)\|_2 \ \geq \ (1 - \varepsilon)b\,\Delta_r.$$

If the margin condition equation 26 holds, these ranges are disjoint, so any $\widetilde{\Delta} \in \big((1 + \varepsilon)a\,\Delta_r,\ (1 - \varepsilon)b\,\Delta_r\big)$ separates the two classes, and the thresholding step in Algorithm 1 succeeds. The theorem follows by taking $\widetilde{\Delta} = \Delta_r$ when $(1 + \varepsilon)a \leq 1 \leq (1 - \varepsilon)b$. $\qquad\square$

