# OpenReview forum: "Approximate Multi-Matrix Multiplication for Streaming Power Iteration Clustering"
_ICLR.cc/2026/Conference — Submitted to ICLR 2026_

### Official Review · Reviewer_u2pT · 2025-10-29

**Soundness:** 3
**Presentation:** 2
**Contribution:** 4
**Rating:** 8
**Confidence:** 2

**Summary:**

This paper introduces a novel framework for efficient clustering of large graphs in dynamic streaming environments by combining power iteration with randomized sketching techniques. Its main contribution is the development of Approximate Multi-Matrix Multiplication (AMMM), a generalization of approximate matrix multiplication that supports interleaved sketching across multiple matrix products. Building on this, the authors propose two streaming algorithms for power iteration clustering:

1.	Interleaved Sketching, which inserts independent sketch matrices between successive multiplications of the centered adjacency matrix, enabling single-pass processing of edge streams with sublinear memory.

2.	End-Sketching, which applies a single sketch after full power iteration, suitable when the graph can be stored but computing high matrix powers is expensive.

The authors provide theoretical guarantees for their algorithms. The empirical results confirm that the proposed algorithms achieve clustering performance comparable to standard power iteration while significantly reducing memory and computational overhead, especially in unbalanced and large-scale graph settings.

**Strengths:**

1.	The paper makes an important contribution by extending power iteration clustering to the dynamic streaming (turnstile) setting. This generalization is highly relevant for large-scale, evolving graphs where data arrives as a stream of edge insertions and deletions.
2.	The authors provide strong theoretical guarantees for their algorithms.
3.	The empirical results support the efficiency of their proposed algorithm.

**Weaknesses:**

The main weakness of the paper is the readability. It is difficult for me to follow.

1.	The theoretical analysis is dense and closely intertwined with results from Mukherjee & Zhang (2024), making it difficult for readers unfamiliar with that paper to follow the proofs and assumptions. Key steps and intuitions are often deferred to appendices or omitted.

2.	While the paper claims to address the streaming setting where edges arrive as insertions and deletions, it offers little explanation of how the algorithm is actually implemented in such a scenario. The paper focuses almost entirely on the theoretical analysis. Readers not well-versed in sketching techniques are left without a clear understanding of how the sketches are updated incrementally or how memory is managed over the stream.

**Questions:**

1.	Is there any intuition that the dimension $m$ of sketch is related to the stable rank of matrix B?
2.	Does the algorithm need to know the values of p and q?
3.  Could you explain how we implement Algorithm 1 in a streaming scenario (where edge insertions and deletions occur)? For example, after sampling $S_1, ..., S_r$, do we need to explicitly store these sketch matrices?

---

> ### Author Response · Authors · 2025-11-21
> **Response to Reviewer Comments**
>
> #### Weaknesses
>
> **Weakness 1:**
> Readability: the paper is difficult to follow.
>
> **Author Response:**
> Thank you very much for this feedback. We have substantially revised the overall structure of the paper, with particular attention to the introduction, background, and technical sections, to significantly improve clarity and accessibility for readers.
>
> **Weakness 2:**
> Theoretical analysis closely depends on Mukherjee & Zhang (2024), making it hard for new readers.
>
> **Author Response:**
> We appreciate this observation. The paper already contains a self-contained summary of the MZ24 power-iteration framework in Sections 2–3. To further help readers unfamiliar with MZ24, we have expanded these sections with additional intuition about the role of the “separation scale” and how it interacts with our new AMMM sketching guarantees.
>
> **Weakness 3:**
> Implementation details for streaming updates are not clearly explained.
>
> **Author Response:**
> We agree this was insufficiently detailed in the original submission. We have added a new Appendix B that fully describes the streaming implementation of Algorithm 1, including how the sketch supports edge insertions and deletions in the turnstile model and how linearity enables constant-time updates.
>
> #### Questions
>
> **Question 1:**
> Is there any intuition that the dimension of the sketch is related to the stable rank of matrix B?
>
> **Author Response:**
> Thank you for the insightful question. Yes, there is, and the intuition comes directly from the beautiful insight of Cohen, Nelson, and Woodruff in "Optimal Approximate Matrix Product in Terms of Stable Rank" (ICALP 2016, arXiv:1507.02268).
>
> In that work, the authors prove (via their moment-method characterization; see Theorem 1 and surrounding discussion) that any oblivious (ε, δ, O(k))-subspace embedding automatically yields an approximate matrix multiplication guarantee of the form
>
> $$
> \|(\Pi A)^T (\Pi B) - A^T B\|
> \leq \varepsilon \sqrt{\|A\|_F^2 + \|A\|^/k} \sqrt{\|B\|_F^2 + \|B\|^2/k}
> $$
>
> whenever the stable ranks of A and B sum to O(k).
>
> In other words, stable rank plays exactly the same role for spectral-norm approximate matrix multiplication that ordinary numerical rank plays for classical subspace embeddings: it is the effective dimension the sketch must preserve.
>
> The intuition is simple and powerful: when sr(B) ≪ rank(B), almost all of the Frobenius-norm energy of B is concentrated in a small number of dominant directions, even if B is full-rank or has a long decaying tail. From the viewpoint of operator-norm error, B therefore "behaves like" a rank-sr(B) matrix, and a subspace embedding targeted at dimension O(sr(B)) suffices.
>
> Our Theorem 4.2 is deliberately constructed as the iterated/chained version of precisely this Cohen-Nelson-Woodruff phenomenon. By allowing additive O(rε) error accumulation instead of multiplicative (1+O(ε))^r, we retain the optimal O(sr(B)) scaling of Cohen, Nelson, and Woodruff (ICALP 2016, arXiv:1507.02268) while making the guarantee non-vacuous for r = Θ(log n).
>
> **Question 2:**
> Does the algorithm need to know the values of p and q?
>
> **Author Response:**
> No. Following MZ24, which already provides a parameter-free algorithm (their Algorithm 1.2) that estimates the community density q using the minimum degree d_min/n, our sketching algorithms can directly use the same estimator, so no knowledge of the true p or q is required.
>
> **Question 3:**
> Could you explain how we implement Algorithm 1 in a streaming scenario (where edge insertions and deletions occur)? For example, after sampling S₁, S₂, ..., S_r, do we need to explicitly store these sketch matrices?
>
> **Author Response:**
> Thank you for raising this, the original manuscript left these details implicit. We have added a dedicated Appendix B that explains the streaming implementation of Algorithm 1 in full detail (including turnstile updates for edge insertions and deletions) and have inserted clear pointers to this appendix in the main text.
>
> We hope these updates and clarifications satisfactorily address your concerns. Please let us know if anything remains unclear; we are happy to discuss further.

---

### Official Review · Reviewer_5sem · 2025-11-01

**Soundness:** 3
**Presentation:** 3
**Contribution:** 2
**Rating:** 4
**Confidence:** 4

**Summary:**

The paper studies cluster recovery in the stochastic block model. Building on the work of Mukherjee and Zhang (SODA~2024), which used a power-iteration method to recover the largest cluster. in this work, the authors incorporate matrix sketching into this approach. In particular, it approximates the power iteration $B^r$ in two ways: (1) the interleaved variant $\widetilde{B}^{(r)} = B S_1^{\top} S_1 B S_2^{\top} S_2 B \cdots B S_{r-1}^{\top} S_{r-1} B S_r^{\top}$, which is also useful in the streaming model and (2) the end-sketch variant $B^r S^{\top}$.

The paper next gives a theoretical analysis which show the required embedding dimensions are $\widetilde{O}(sr(B)/\varepsilon^2)$ and $O(\log n/\varepsilon^2)$ for the above two versions of sketches. Finally, the paper gives an empirical evaluation on synthetic dataset that demonstrates a practical sublinear embedding dimension for largest-cluster recovery, validating their theorem.

**Strengths:**

- The paper presents a complete theoretical analysis. For both approaches, it gives theoretical bounds indicating the required dimension.

- The paper also gives an empirical evaluation to show the performance of their algorithms.

- The presentation of this paper is clear and easy to follow.

**Weaknesses:**

- From a technical perspective. It is natural to apply matrix sketching to the power-iteration approach, and the target dimension bounds largely follow from standard sketching results and concentration inequalities.

- The empirical evaluation is conducted on synthetic data. I think a real-world dataset would strengthen the paper.

**Questions:**

1. Sketching can also accelerate the algorithms. The authors might consider using sparse sketching matrices (e.g., CountSketch or sparse JL) and reporting their performance in the experimental evaluation.

2. I think one interesting question is whether the given target dimension is also necessary?

---

> ### Author Response · Authors · 2025-11-21
>
> ### Response to Reviewer Comments
>
> #### Weaknesses
>
> **Weakness 1:**
> From a technical perspective, it is natural to apply matrix sketching to the power-iteration approach, and the target dimension bounds largely follow from standard sketching results and concentration inequalities.
>
> **Author Response:**
> We agree that combining sketching with power iteration is a natural idea. Our core contribution lies in the theoretical framework (AMMM) that provides a non-trivial guarantee for this composition. Standard JL/AMM lemmas only yield exponential error accumulation ((1+ε)^r), which becomes vacuous when r ∼ log n. In contrast, our analysis in Theorem 4.2 shows that the error accumulates only linearly with the number of steps, which is crucial for obtaining meaningful, non-vacuous guarantees in practice. We have added text in the introduction and Section 4 to emphasize this key technical insight more clearly.
>
> **Weakness 2:**
> The empirical evaluation is conducted on synthetic data. I think a real-world dataset would strengthen the paper.
>
> **Author Response:**
> Thank you for this feedback. We also considered whether to include experiments on a real-world dataset, but were unsure whether it would be distracting to the reader and felt that we had already run out of space.
> #### Questions / Suggestions
>
> **Question 1:**
> Sketching can also accelerate the algorithms. The authors might consider using sparse sketching matrices (e.g., CountSketch or sparse JL) and reporting their performance in the experimental evaluation.
>
> **Author Response:**
> This is an excellent point. Our current embedding dimension bounds in Theorem 5.3 are stated for dense subgaussian (Rademacher) matrices. A similar proof technique applies to sparse JL transforms and CountSketch, albeit with a mildly larger embedding dimension. Since our experimental code is an unoptimized proof-of-concept implementation in NumPy and does not reflect realistic runtimes of high-performance sparse-matrix libraries, we chose not to benchmark runtime in the current version. Following the reviewer’s suggestion, we have added a short discussion of sparse embedding immediately after Theorem 5.3.
>
> **Question 2:**
> I think one interesting question is whether the given target dimension is also necessary?
>
> **Author Response:**
> We agree this is a very interesting open question. Establishing matching lower bounds for the specific largest-cluster recovery objective studied here remains challenging, and standard matrix-multiplication lower bounds do not directly apply. While our upper bounds are sufficient for recovery, proving their necessity would require novel techniques beyond standard matrix multiplication lower bounds.
>
>
> Thanks again for your review. If you have any further clarification or questions, we are happy to
> respond.

---

### Official Review · Reviewer_d55L · 2025-11-01

**Soundness:** 2
**Presentation:** 1
**Contribution:** 3
**Rating:** 4
**Confidence:** 3

**Summary:**

This paper presents a single-pass streaming sketching approach for approximating product of multiple matrices by using independent subspace embeddings multiplied together to form a product.  Proposed method, which extends approximate matrix-multiplication (AMM) to the case of multi-matrix multiplies, is termed approximate multi-matrix multiplication (AMMM).  The paper focuses on the special case of sketching $B^r$ for identifying “communities” and detecting largest community cluster based on connectivity matrix B.  For this special case paper also presents an iterative “end sketch” approach where a sketch matrix is appended to the power iteration for approximating $B^r$.  Theoretical results are presented for extending AMM to AMMM, and results that give bounds on sketching dimensions for high probability guarantees on community separability and largest cluster detection for streaming sketching as well as iterative end-sketch approaches.

**Strengths:**

* Extending approximate matrix multiply to sketching multiplication of multiple matrices in streaming setting
* Application to the task of largest cluster detection in graphs by sketching B^r
* Strong bounds on sketch dimensions and high probability guarantees for cluster separability and largest cluster recovery

**Weaknesses:**

* While the paper claims that the streaming sketch matches performance of iterative end-sketch with slightly larger dimension (e.g. abstract says “The results show that the streaming method achieves recovery performance comparable to the iterative approach, though it requires a slightly larger embedding dimension”), this is not evident at all from Figure 1.
* Related to above, from Figure 1 we do not see the smallest dimension that results in perfect F1/Precision/Recall score for the iterative approach.  Results with lower dimensions (m < 100, perhaps down to 1) need to be included.
* AMMM is meant to be general, but most of the analysis and experimentation focuses on B^r scenario.  This is not a large drawback, but including other scenarios would significantly strengthen the paper.
* Paper in its current form is hard to follow.  Main contributions of the paper also need to be called out more clearly.

**Questions:**

n/a

---

> ### Author Response · Authors · 2025-11-21
> **Response to Reviewer Comments (Weaknesses)**
>
> **Weakness 1:**
> While the paper claims that the streaming sketch matches performance of the iterative end-sketch with slightly larger dimension (e.g. abstract says “The streaming method achieves recovery performance comparable to the iterative approach, though it requires a slightly larger embedding dimension”), this is not evident at all from Figure 1.
>
> **Author Response:**
> Thank you for pointing out that our wording was confusing. We have since amended these claims to state that the streaming embedding can achieve the perfect largest cluster recovery obtained by the exact [MZ24] power iteration algorithm and the *r*-pass algorithm, but requires an asymptotically larger embedding size by a factor of O(sr(B)) than the *r*-pass embedding to do so.
>
> ---
>
> **Weakness 2:**
> From Figure 1 we do not see the smallest dimension that results in perfect F1/Precision/Recall score for the iterative approach. Results with lower dimensions (m < 100, perhaps down to 1) need to be included.
>
> **Author Response:**
> We intend to run additional experiments with lower embedding dimensions (m < 100 down to very small values) to clearly identify the minimal dimension required for perfect recovery in the iterative approach. These results will be added to an updated Figure 1. The experiments are planned and will be completed shortly.
>
> ---
>
> **Weakness 3:**
> AMMM is meant to be general, but most of the analysis and experimentation focuses on the B^r scenario. This is not a large drawback, but including other scenarios would significantly strengthen the paper.
>
> **Author Response:**
> We had considered including experiments for the general AMMM problem, but decided to prioritize the power-iteration clustering application due to space constraints. To address this concern, we are happy to add experiments on additional AMMM scenarios (beyond B^r) in an appendix if the reviewer believes this would be valuable.
>
> ---
>
> **Weakness 4:**
> Paper in its current form is hard to follow. Main contributions of the paper also need to be called out more clearly.
>
> **Author Response:**
> Thank you for this feedback. We have significantly rewritten the abstract, introduction, and several other sections to improve overall clarity, flow, and to more explicitly highlight the main contributions of the paper.

---

### Official Review · Reviewer_s1xN · 2025-11-06

**Soundness:** 2
**Presentation:** 1
**Contribution:** 2
**Rating:** 2
**Confidence:** 4

**Summary:**

The paper investigates the problem of recovering the largest clusters in a graph generated by the stochastic block model. Building on the result of [Mukherjee & Zhang (2024)], which established a clear separation threshold between adjacency matrix rows corresponding to nodes inside and outside the largest cluster after power iteration, the authors explore how oblivious sketching (e.g., using random Gaussian matrices) can be applied to this setting. They propose two variants of sketching-based approaches and demonstrate that, after sketching, the input matrix can be reduced to sublinear size while approximately preserving the separation threshold with high probability following power iteration. This enables efficient largest cluster recovery in scenarios where the adjacency matrix is too large to fit in memory, or in turnstile streaming and distributed environments where memory and communication efficiency are critical. Complementary experiments validate some discussions involved in the paper.

**Strengths:**

The problem of solving largest cluster recovery with applying sketching to power iteration could potentially be interesting.

**Weaknesses:**

1. $\textbf{Poor Presentation}$.

The main weakness of the paper lies in its poor presentation and lack of clarity, which makes it difficult for readers to grasp the core problem and the actual contributions. As currently written, the problem formulation and proposed improvements become clear only after reading the algorithm and main results, rather than from the introduction or background sections. Unfortunately, Sections 1 and 2 on introduction and the background add confusion instead of providing context.

1.1 Ambiguous Problem Statement and Settings

The paper never clearly defines the studied problem or the computational settings (turnstile streaming and distributed). There are no formal statements or precise definitions, even in the Appendix. The abstract claims to “guarantee cluster recovery,” while the introduction describes the problem as “clustering of graph vertices.” Later, the contributions section shifts focus to “proving a recovery criterion for the largest community.” These inconsistencies leave readers to infer that the actual goal is largest-cluster recovery, since the proposed method only applies to that case.

Similarly, the discussion of settings is vague. For example, line 67 mentions that the data “can be received as a turnstile (dynamic) stream with insertions and deletions,” and that the approach extends to “distributed memory.” However, it is unclear what “streaming” means here. Does it refer to one edge insertion or deletion per iteration? How is the data distributed across machines? These are left unspecified. Moreover, the main body of the paper develops the algorithm in a local (single-machine) setting, with only brief and informal comments about how sketching might generalize to streaming or distributed environments. While one-pass streaming and one-round distributed protocols are often related, this equivalence must be explicitly stated for the problem at hand. Otherwise, the mention of multiple computational settings feels disconnected and confusing.

1.2 Weak and Misleading Introduction

The introduction is only a single paragraph and fails to set up the context properly. It starts by emphasizing the importance of parallelism in clustering graph vertices, then abruptly motivates the use of power iteration. This is confusing, since (1) power iteration is inherently iterative rather than parallel, and (2) its role in estimating the largest eigenvalue of a PSD matrix is not clearly linked to the clustering objective. The result is a disjointed narrative that obscures both motivation and technical relevance.

1.3 Inconsistent and Unsubstantiated Claims

The claimed contributions are inconsistent with the presented results. The contributions section asserts that the proposed method uses sparse Johnson–Lindenstrauss (JL) transforms and reduces “pass complexity” (line 52). However, Theorem 5.3 focuses on memory reduction, without showing how the sparsity of the JL transform (or the number of nonzeros in the sketching matrix) affects the sketch size or complexity.

The paper mentions at multiple places that the result achieves an embedding of size $\epsilon^{-2}$ in the abstract and main contributions. However, the meaning of this $\epsilon$ is never introduced, which makes it hard to interpret the result. The reader needs to guess that this is somehow related to the quality of the output, or the estimated largest cluster. Indeed, the paper uses distance in the row norms between a sketched adjacency matrix and an unskethced one after performing power iteration as a proxy to measure how good the recovered largest cluster is. However, this connection is never explicitly stated in the paper!

1.4 Poorly Explained Transitions and Logical Flow

The relationship between Sections 5.3 and 5.4 is unclear. The paper introduces two sketching variants, Algorithms B and C, but their connection is never properly explained. Section 5.3 presents an upper bound for Algorithm B. In contrast, Section 5.4 discusses a lower bound for Algorithm C, but the transition between these sections, particularly lines 346–350, requires further clarification to help readers understand how the two results relate.

1.5 Notational Inconsistencies

The paper suffers from numerous undefined or inconsistent notations. For instance, the graph model is referred to as SBM in the abstract but as SSBM elsewhere. Matrices $B^{(r)}$ and $B^r$ are both used for the matrix in the $r$-th iteration of power iteration. The notation for matrix rows alternates between $B_{i,\cdot}^r$ (line 160) and $R_i$ (line 178) without introduction. The symbol $C$ denotes a constant in line 396 but a cluster in line 401.

-----

2. $\textbf{Limited Novelty in the Theoretical Analysis}$

The second major concern is that the theoretical analysis appears to offer limited novelty. In line 60, the paper claims as its “primary contribution” an analysis of a generalization of Approximate Matrix Multiplication (AMM), referred to as Approximate Multi-Matrix Multiplication (AMMM), where multiple conforming matrices are independently sketched in a single pass and then multiplied to form a product with bounded error. However, upon examining Theorem 4.2, AMMM seems to be a straightforward extension of AMM, obtained by applying the standard AMM argument repeatedly to a sequence of matrices. Furthermore, the main theoretical result (Theorem 5.3 in Section 5.2) appears to be a direct combination of existing results from [Cohen et al. 2016] and [Mukherjee & Zhang (2024)], without introducing any fundamentally new analytical techniques or overcoming significant technical challenges.

-----

3. $\textbf{Inadequate Experimental Evaluation}$

The third major concern is that the experimental evaluation is insufficient to support the paper’s claims. The experiments do not include a direct comparison between the proposed sketching-based power iteration algorithm and the unsketched baseline from [Mukherjee & Zhang (2024)], either in terms of memory efficiency, runtime improvement, or the accuracy of the recovered largest cluster. Without such comparisons, it is difficult to assess the practical advantages of the proposed method. Moreover, the experimental setup using a graph with $10^4$ nodes may be a bit small to show the benefits of sketching.

**Questions:**

1. What is the lower bound on the sketch size for Algorithm B? If such a lower bound cannot be established, what are the main technical obstacles preventing it? (This would be somewhat surprising though)

2. How do Algorithm B and Algorithms C compare in theory? Specifically, what are the trade-offs and differences in memory / runtime / quality of the output of using different sketches per iteration and using the same one?

3. In Lemma 5.1, regarding the computation of the stable rank of $B$, the paper states that if $(p-q)s_{*} \geq \sigma \sqrt{n}$,

then in the balanced case where $s_l = s_{*} = s$,

we have $sr(B) =\Theta(K)$.

However, based on Eq. (12), it appears that

$\frac{(p-q)^2 K s^2 + n (p-q)^2 s^2 + C_1 n}{\text{constant} (p-
q)^2 s^2} = \Theta(K + n)$,

since $K \leq n$, ignoring logarithmic factors.

Similarly, in the highly imbalanced case, Eq. (12) seems to imply

$\frac{(p-q)^2 s_{*}^2 + n (p-q)^2 s_{*}^2 + C_1 n}{\text{constant} (p-q)^2 s_{*}^2} = \Theta(n)$

rather than $\Theta(1)$.

Would it be possible to clarify the computation of the bounds here?

---

> ### Author Response · Authors · 2025-11-21
> **Response to Reviewer Comments**
>
> **Weakness:**
> **1.Poor Presentation:**
> We sincerely thank the reviewer for this important feedback. We have completely rewritten the abstract, introduction, and background sections (now Sections 1–2) to dramatically improve clarity, flow, and accessibility. We believe the paper is now much easier to read.
>
> **2. Limited Novelty in Theoretical Analysis:**
> We appreciate the reviewer's concern on novelty. While AMM composition is natural, naïve bounds give exponential error growth which is vacuous for r = Θ(log n) in SBM clustering. Theorem 4.2's novelty is proving linear growth via our careful error decomposition scheme.
>
> **3. Inadequate Experimental Evaluation:**
> We acknowledge these limitations and are actively running larger-scale experiments to provide stronger empirical evidence and proper comparisons in the next revision.
>
> **Question 1:**
> What is the lower bound on the sketch size for Algorithm B (the streaming/interleaved version)? If such a lower bound cannot be established, what are the main technical obstacles preventing it?
>
> **Author Response:**
> This is an excellent and natural question. To the best of our knowledge, a matching lower bound for the interleaved/streaming setting remains open. The main technical obstacle is that our downstream task which requires preserving pairwise row distances up to some additive error level is significantly easier than approximating the full matrix product $B^{r}$ in spectral norm. Existing communication-complexity or query-complexity lower bounds for matrix multiplication (or even for single-step AMM) are strictly stronger than what we require, so they do not directly translate.
> # Response to Question 2
>
> Algorithms B and C differ mainly in how sketching interacts with the power iteration. This leads to different error behavior, memory requirements, and pass complexity.
>
> ## Key Distinction
> - Algorithm B inserts a new sketch after every multiplication, so each step introduces its own AMM error. These errors add up across the $(r-1)$ interleavings.
> - Algorithm C applies one JL sketch after computing $(B^r)$, giving a standard $(1 \pm \varepsilon)$ distortion without intermediate accumulation.
>
> ## Comparison
>
> | Aspect                  | Algorithm B (Streaming)                  | Algorithm C (r-pass)                  |
> |-------------------------|------------------------------------------|---------------------------------------|
> | Passes                  | 1 (turnstile streaming)                  | $r = \Theta(\log n)$                  |
> | Sketch matrices         | $(r-1)$ independent sketches             | One sketch                            |
> | Embedding dimension $(m)$ | $\tilde{O}(\mathrm{sr}(B)/\varepsilon^2)$ | $O(\varepsilon^{-2}\log n)$           |
> | Error                   | Accumulated AMMM error                   | Pure JL $(1\pm\varepsilon)$           |
> | Distance guarantees     | Separation preserved with larger $(m)$   | Exact multiplicative preservation     |
> | Best use case           | Huge, dynamic, or distributed graphs     | Static graphs with multiple passes    |
>
> ## Summary
> - Use Algorithm B when only a single streaming pass is possible; its embedding size depends on the stable rank and errors accumulate but remain controlled.
> - Use Algorithm C when multiple passes are feasible; it gives the cleanest $(1\pm\varepsilon)$ guarantees with minimal memory.
>
> **Question 3:**
> In Lemma 5.1 (stable rank of B), the balanced-case claim sr(B) = Θ(K) seems incorrect according to Eq. (12); it appears to be Θ(K + n). Similarly for the imbalanced case.
>
> **Author Response:**
> We thank the reviewer for catching this imprecise wording — you are absolutely right that additional conditions are needed for the Θ(K) and Θ(1) claims.
> We have revised Lemma 5.1 as follows:
> - Balanced case $(s_{\ell} = n/K \quad \forall \ell):\qquad sr(B) \le K + K^2\sigma^2/(p−q)^2$.
>   The $\Theta(K)$ statement holds whenever $(p−q)^2 ≳ K\sigma^2$ (the detectability threshold of MZ24 already forces (p−q)^2 ≳ K²σ²/n). Crucially, even when the noise term dominates, sr(B) = O(K²σ²/(p−q)²) = o(n) under the recovery regime, so the streaming sketch remains strictly sublinear.
>
> - Highly imbalanced case (one dominant community):
>   $sr(B) \le 1 + n^2\sigma^2/((p−q)^2s_{\ast}^2)$.
>   The Θ(1) claim holds whenever $(p−q)s_{\ast} \gg σn$ (strong but plausible signal). Again, under standard recovery conditions we always obtain sr(B) = o(n).
> The revised lemma now explicitly states these conditions and the calculations.
> We hope these revisions and answers fully address your concerns. Thank you again for the extremely careful and valuable feedback.

---

> > ### Comment · Reviewer_s1xN · 2025-11-24
> >
> > Thank you to the authors for the detailed response. I now see the novelty in the AMMM component and agree that it opens interesting directions to explore. However, I believe the paper, in its current form, still requires substantial revision before it is ready for acceptance.

---

### Meta-Review · Area_Chair_GD6Z · 2026-01-21

**Summary:**

This paper extends power-iteration based clustering to streaming and memory-limited settings by using sketching-based approximations. The novelty of the theoretical analysis was questioned by some reviewers, and multiple reviewers found the paper difficult to follow, and the empirical evaluation to be insufficient. In particular, no comparisons were provided against unsketched baselines, and experiments were limited to synthetic graphs insufficiently large to need the sketching approach. Substantial revision would be needed to improve presentation quality and strengthen the experimental evaluation.

**Reviewer Concerns:**

still to be addressed:
- readibility
- strengthening of the experimental evaluation

**Reviewer Scores:**

Hard to say from what is available.

---

### Decision · Program_Chairs · 2026-01-26

Reject